# Modulation of Adhesion Process, E-Selectin and VEGF Production by Anthocyanins and Their Metabolites in an In Vitro Model of Atherosclerosis

**DOI:** 10.3390/nu12030655

**Published:** 2020-02-28

**Authors:** Mirko Marino, Cristian Del Bo’, Massimiliano Tucci, Dorothy Klimis-Zacas, Patrizia Riso, Marisa Porrini

**Affiliations:** 1Division of Human Nutrition, Department of Food, Environmental and Nutritional Sciences, Università degli Studi di Milano, 20133 Milan, Italy; mirko.marino@unimi.it (M.M.); massimiliano.tucci@unimi.it (M.T.); patrizia.riso@unimi.it (P.R.); marisa.porrini@unimi.it (M.P.); 2School of Food and Agriculture, University of Maine, Orono, ME 04469, USA; dorothea@maine.edu

**Keywords:** anthocyanins and metabolites, inflammation, adhesion molecules, vascular endothelial growth factor, monocytes, endothelial cells

## Abstract

The present study aims to evaluate the ability of peonidin and petunidin-3-glucoside (Peo-3-glc and Pet-3-glc) and their metabolites (vanillic acid; VA and methyl-gallic acid; MetGA), to prevent monocyte (THP-1) adhesion to endothelial cells (HUVECs), and to reduce the production of vascular cell adhesion molecule (VCAM)-1, E-selectin and vascular endothelial growth factor (VEGF) in a stimulated pro-inflammatory environment, a pivotal step of atherogenesis. Tumor necrosis factor-α (TNF-α; 100 ng mL^−1^) was used to stimulate the adhesion of labelled monocytes (THP-1) to endothelial cells (HUVECs). Successively, different concentrations of Peo-3-glc and Pet-3-glc (0.02 µM, 0.2 µM, 2 µM and 20 µM), VA and MetGA (0.05 µM, 0.5 µM, 5 µM and 50 µM) were tested. After 24 h, VCAM-1, E-selectin and VEGF were quantified by ELISA, while the adhesion process was measured spectrophotometrically. Peo-3-glc and Pet-3-glc (from 0.02 µM to 20 µM) significantly (*p* < 0.0001) decreased THP-1 adhesion to HUVECs at all concentrations (−37%, −24%, −30% and −47% for Peo-3-glc; −37%, −33%, −33% and −45% for Pet-3-glc). VA, but not MetGA, reduced the adhesion process at 50 µM (−21%; *p* < 0.001). At the same concentrations, a significant (*p* < 0.0001) reduction of E-selectin, but not VCAM-1, was documented. In addition, anthocyanins and their metabolites significantly decreased (*p* < 0.001) VEGF production. The present findings suggest that while Peo-3-glc and Pet-3-glc (but not their metabolites) reduced monocyte adhesion to endothelial cells through suppression of E-selectin production, VEGF production was reduced by both anthocyanins and their metabolites, suggesting a role in the regulation of angiogenesis.

## 1. Introduction

Inflammation represents the initial response of the body to harmful stimuli (i.e., pathogens, injury) and involves the release of numerous substances known as inflammatory mediators. Normally, inflammatory stimuli may activate intracellular signaling pathways that promote the production of inflammatory mediators including microbial products (i.e., lipopolysaccharide) and cytokines such as interleukin-1β (IL-1β), interleukin-6 (IL-6), and tumor necrosis factor-α (TNF-α). However, the inflammatory response also involves the activation of cells such as macrophages and monocytes that are able to mediate local responses resulting from tissue damage and infection [1]. In particular, activated endothelial cells release numerous cell surface adhesion molecules such as vascular cell adhesion molecule (VCAM)-1, intra-cellular adhesion molecule (ICAM)-1, P-selectin and E-selectin (also known as the endothelial leucocyte adhesion molecule—ELAM), which attract neutrophils and monocytes at the endothelial level, permit their adhesion and transmigration into the tissue and increase microvascular permeability [2,3]. Generally, inflammation is of relatively short duration. When uncontrolled, inflammation becomes chronic and can contribute to the pathogenesis of many diseases, including chronic inflammatory diseases and degenerative diseases such as atherosclerosis.

Inflammation may also promote angiogenesis, a process that involves the formation of new blood vessels from preexisting ones. Angiogenesis is associated with the activation and proliferation of endothelial cells, and structural changes of the vasculature. Vascular endothelial growth factor (VEGF) is important for endothelial integrity, vascular function and angiogenesis. In fact, VEGF can stimulate endothelial cell survival, invasion and migration into surrounding tissues and increase proliferation and vascular permeability. On the other hand, during atherosclerosis, VEGF may enhance the pathophysiologic mechanism of plaque formation and destabilization by increasing the risk of plaque rupture [4,5].

Polyphenols are a heterogeneous class of bioactive compounds found abundantly in the plant kingdom. They are generally classified into phenolic acids (hydroxycinnamic and hydroxybenzoic acids), flavonoids (flavanols, flavonols, flavons, flavanones, isoflavons and anthocyanidins), stilbens and lignans. Polyphenols are responsible for the color, bitterness, astringency, flavor and smell of numerous plants including fruits, vegetables, coffee, chocolate and tea [6]. In foods, most of them are present as glycosides. After ingestion, polyphenols move intact through the gastrointestinal tract to the small intestine where they are absorbed through passive (i.e., aglycones) and/or active transport (i.e., glycosides). Polyphenols that enter intestinal epithelial cells are metabolized in the intestine and liver through methylation, glucuronidation and sulfation reactions [7,8]. Unabsorbed polyphenols reach the colon where they are extensively metabolized by gut microbiota [9]. Additionally, the microbial derivatives after absorption undergo conjugation and are metabolized in the liver. Polyphenols reach maximal plasma concentration within 1.5 h after absorption and disappear from the bloodstream by 6 h post-consumption, while their metabolites may display a biphasic phase (depending on microbiota and endogenous metabolism) and appear in the blood 8–10 h or 16–24 h after consumption [10,11]. It is estimated that plasma concentrations range between nanomolar for anthocyanins and other polyphenols (native forms), up to low micromolar for their derivates [10,11]. 

In recent years, polyphenols have received extensive interest for their health benefits in the prevention of numerous cardiovascular diseases [12,13,14,15,16]. The mechanisms through which polyphenols may exert their bioactivity are not completely understood since it is not clear whether their activity is linked to the native forms, their derivatives or both. Some of the most proposed protective mechanisms of action include the increase of antioxidant/detoxification enzymes activity (i.e., glutathione S-transferase, superoxide dismutase, glutathione peroxidase) [17,18,19], and the decrease of pro-inflammatory cytokines (i.e., tumor necrosis factor alpha (TNF-α), interleukin-1, interleukin-6) [20,21,22]. Furthermore, polyphenols have been documented to have the capability to modulate some factors involved in atherosclerosis, such as the release of numerous vasoconstrictor and vasodilator agents at the endothelial level including nitric oxide, endothelin-1 and soluble vascular cell adhesion molecules-1 (sVCAM-1) [23]. In this regard, we have previously reported the ability of different anthocyanins and metabolites to counteract and/or resolve an inflammation-driven adhesion of monocytes on endothelial cells (HUVECs). In the present study, we focused on the effects of peonidin (peo) and petunidin (pet)-3-glucoside, two anthocyanins mainly found in berries and grapes [24], and their respective metabolites (vanillic and methyl-gallic acids; VA and MetGA) on their capacity to resolve a TNF-α-mediated inflammatory process responsible for the adhesion of monocytes to HUVECs through the production of the mediators VCAM-1 and E-selectin. In addition, since TNF-α and monocytes play a crucial role in angiogenesis [25], we evaluated whether polyphenolic compounds were also able to reduce VEGF production, one of the main angiogenic factors. To the best of our knowledge, very few studies have explored this topic, as the majority of them focus on oncology.

## 2. Materials and Methods 

### 2.1. Chemicals and Reagents

Lyophilized standards of peonidin-3-glc (Peo-3-glc) and petunidin-3-glucoside (Pt-3-glc) were purchased from Polyphenols Laboratory (Sandes, Norway). Lyophilized standards of vanillic acid (VA) and methil-gallic acid (MetGA), Hanks balanced salt solution, fetal bovine serum (FBS), tumor necrosis factor-alpha (TNF-α), MTT kit, Trypan blue and Triton X-100 were obtained from Sigma-Aldrich (St. Louis, MO, USA). Sodium pyruvate, RPMI-1640, HEPES, gentamicin and trypsin–EDTA (0.05%) and gelatine (0.1%) were from Life Technologies (Monza Brianza, MB, Italy). Human endothelial cell basal medium and the growth supplement were obtained from Tebu-Bio (Magenta, MI, Italy), while 5-chloromethylfluorescein diacetate (CellTracker^TM^ Green; CMFDA) was obtained from Invitrogen (Carlsbad, CA, USA). Methanol and hydrochloric acid (37%) were obtained from Merck (Darmstadt, Germany), while water from a Milli-Q apparatus (Millipore, Milford, MA) was used. VCAM-1 and VEGF ELISA kits were purchased from Vinci-Biochem Srl (Vinci, FI, Italy) and the E-Selectin ELISA kit was purchased from Aurogene Srl (Roma, RM, Italy). 

### 2.2. Preparation of Anthocyanin and Metabolite Standards

The stock solutions of Peo-3-glc, Pet-3-glc, VA and MetGA (Figure 1) were prepared by dissolving the powder of each standard in acidified methanol (0.05% HCl). Successively, standards were quantified spectrophotometrically and stored in dark vials at −80 °C until use.

### 2.3. Cell Culture

Monocytic cells (THP-1; Sigma-Aldrich, St. Louis, MO, USA) were cultured in complete RPMI cell medium (RPMI-1640 medium supplemented with 1% HEPES, 1% sodium pyruvate, 0.1% gentamicin and 10% FBS). For the experiment, 1 × 10^5^ cells were grown in a flask until the concentration of 1 × 10^6^ cells/mL was reached. Human umbilical vein endothelial cells (HUVECs; Tebu-Bio Srl, Magenta, MI, Italy) were seeded at the concentration of 1 × 10^5^ cells on a pre-coated flask with 0.1% gelatine and grown in a cell medium kit containing 2% serum until they reached confluence.

### 2.4. Cytotoxicity Assay

The cytotoxicity of the compounds was tested by Trypan blue and (3-(4,5-dimethylthiazol-2-yl)-2, 5-diphenyltetrazolium bromide (MTT) assay on HUVECs, according to the manufacturer’s instructions. Triton X-100 was used as positive control. Two independent experiments were performed in which each compound and concentration was tested in quadruplicate.

### 2.5. Evaluation of Monocytes Adhesion on Activated Human Umbilical Vein Endothelial Cells

When the confluency reached about 80%, HUVECs were removed using trypsin (0.05 mM) and seeded on a 0.1% gelatin pre-coated 96-well black plate at the concentration of 2 × 10^4^ cells/well at 37 °C and 5% CO_2_. After 24 h incubation, THP-1 (2 × 10^6^) cells were labelled with CellTracker^TM^ Green CMFDA (1 µM) in 1 mL serum-free RPMI medium (containing 1% HEPES, 1% sodium pyruvate and 0.1% gentamicin) for 30 min. Successively, cells were washed twice, re-suspended in HUVEC medium at a final concentration of 2 × 10^5^ cells mL^−1^ and added to HUVECs. The adhesion process was induced for 24 h with 100 ng mL^−1^ of TNF-α. Then, 200 μL of new medium containing the single compounds (0.02 µM, 0.2 µM, 2 µM and 20 µM for Peo and Pet-3-glc and 0.05 µM, 0.5 µM, 5 µM and 50 µM for VA and MetGA) was added and the cells were further incubated for 24 h. Medium from each well was collected and stored at −80 °C until ELISA analysis. Cells were rinsed twice with 200 µL of Hanks balanced salt solution and the fluorescence (excitation: 485 nm, emission: 538 nm) associated with the number of labeled THP-1 cells attached to the HUVECs was measured by a spectrophotometer (mod. F200 Infinite, TECAN Milan, Italy). Each compound and concentration were tested in quintuplicate in three independent experiments.

### 2.6. ELISA Quantification of Soluble VCAM-1, E-Selectin and VEGF

At the end of the experiment, the recovered cell culture supernatants were used to quantify the concentrations of soluble VCAM-1 (Cat# EK0537, BosterBio), E-selectin (Cat# MBS355367, MyBioSource) and VEGF (Cat# V3-200-820-VEF, Vinci-Biochem). The analysis was performed using ELISA kits according to the manufacturer’s instruction. The analyses were conducted in quadruplicate and the results derived from three independent experiments.

### 2.7. Data Analysis 

STATISTICA software (StatSoft Inc., Tulsa, OK, USA) was used for the statistical analysis. All the results are expressed as means ± standard error of the mean (SEM). One-way ANOVA was applied to verify the effect of Peo-3-glc, Pet-3-glc, VA and MetGA supplementation on cell cytotoxicity, adhesion process and production of soluble VCAM-1, E-selectin and VEGF. The least significant difference (LSD) test was used to assess differences between treatments by setting the level of statistical significance at *p* < 0.05.

## 3. Results

### 3.1. Effect of Peo-3-glc, Pet-3-glc, VA and MetGA on Cell Cytotoxicity

Table 1 presents the effects of the compounds tested on cell cytotoxicity measured by Trypan blue assay at all concentrations tested. Peo-3-glc and Pet-3-glc (from 0.02 µM to 20 μM), VA and MetGA (from 0.05 µM to 50 μM) did not have any cytotoxic effect, maintaining cell viability above 90%. The results were also in line with those obtained following the MTT assay tested only at the maximum concentration (20 μM for anthocyanins (ACNs) and 50 μM for metabolites). Conversely, incubation of HUVEC cells with Triton X-100, as a positive control (data not shown), significantly reduced (*p* < 0.0001) cell viability up to 20% compared to the cells treated with and without TNF-α (cell viability at 99%).

### 3.2. Effect of Peo-3-glc, Pet-3-glc, VA and MetGA on THP-1 Adhesion to HUVECs 

The results of THP-1 adhesion to HUVECs after incubation with Peo-3-glc and Pet-3-glc are shown in Figure 2. Data on the adhesion process are reported as fold increase compared to the control cells without TNF-α or (poly)phenolic compounds. Stimulation with 100 ng mL^−1^ of TNF-α significantly increased (*p* < 0.0001) the adhesion process of THP-1 cells to HUVECs compared to the negative control (no TNF-α). The treatment with Peo-3-glc and Pet-3-glc significantly decreased the (*p* < 0.0001) adhesion of monocytes to HUVECs compared to the TNF-α treatment. The size of the effect was similar between Peo-3-glc (−37%, −24%, −30% and −47%; Figure 2a) and Pet-3-glc (−37%, −33%, −33% and −45%; Figure 2b) at all the concentrations tested (0.02 µM, 0.2 µM, 2 µM and 20 μM, respectively). Figure 3 shows the results of THP-1 adhesion to HUVECs after incubation with VA and MetGA (metabolites of Peo-3-glc and Pet-3-glc, respectively). Only VA (Figure 3a) significantly reduced the adhesion process at the concentration of 50 μM (−21%; *p* < 0.001), while no effect was observed for MetGA (Figure 3b).

### 3.3. Effect of Peo-3-glc, Pet-3-glc, VA and MetGA on the Levels of E-Selectin

Table 2 reports the levels of E-selectin quantified in the cell supernatant following incubation with ACNs and metabolites. Cell stimulation with TNF-α significantly increased (*p* < 0.001) the levels of E-selectin compared to the negative control (without TNF-α). The incubation with Peo-3-glc and Pet-3-glc significantly attenuated (*p* < 0.001) the production of E-selectin. The size of the effect was similar between Peo-3-glc (−55%, −66%, −65% and −76%) and Pet-3-glc (−64%, −60%, −67% and −72%) at all the concentrations tested (0.02 μM, 0.2 μM, 2 μM and 20 μM, respectively). In addition, Peo-3-glc at the high doses (0.2 μM, 2 μM and 20 µM) significantly reduced (*p* < 0.05) the levels of E-selectin (−32%, −31% and −53%, respectively) compared to the negative control (without TNF-α). A similar effect was documented for Pet-3-glc which showed a reduction (*p* < 0.05) at low (0.02 µM; −28%) and high doses (2 μM and 20 µM; −36% and −45%, respectively).

Vanillic acid decreased E-selectin levels at the high dose (50 μM) with respect to the positive TNF-α control (−70%; *p* < 0.001) and the negative control without TNF-α (−46%; *p* < 0.05). Conversely, no effect was observed following MetGA exposure.

### 3.4. Effect of Peo-3-glc, Pet-3-glc, VA and MetGA on the Levels of Soluble VCAM-1

Table 3 presents the levels of VCAM-1 quantified in the cell supernatant following incubation with ACNs and metabolites. Cell stimulation with TNF-α significantly increased (*p* < 0.001) the levels of VCAM-1 compared to the negative control (without TNF-α). Incubation with Peo-3-glc significantly reduced (*p* < 0.0001) the levels of soluble VCAM-1 (−195%, −203%, −69% and −112%) at all concentrations tested (0.02 μM, 0.2 μM, 2 μM and 20 μM, respectively) with maximum reduction at the low doses. Pet-3-glc attenuated soluble VCAM-1 production only at the maximum dose (−270%; 20 μM, *p* < 0.0001) while VA and MetGA had no effect.

### 3.5. Effect of Peo-3-glc, Pet-3-glc, VA and MetGA on the Levels of VEGF

In Table 4, the levels of VEGF quantified in the cell supernatant following incubation with ACNs and metabolites are reported. Cell stimulation with TNF-α induced a small but significant increase (*p* < 0.01) in VEGF levels compared to the negative control (without TNF-α). Incubation with Peo-3-glc and Pet-3-glc significantly reduced (*p* < 0.001) VEGF concentrations. The size of the effect was similar between Peo-3-glc (−27%, −28%, −30% and −30%) and Pet-3-glc (−24%, −27%, −28% and −30%) at all concentrations tested (0.02 μM, 0.2 μM, 2 μM and 20 μM, respectively) and comparable to the negative control (*p* > 0.05). A reduction was also reported for VA (−12%; −17%, −13% and −21%) and MetGA (−9%; −17%, −17% and −19%) at all concentrations tested (0.05 μM, 0.5 μM, 5 μM and 50 μM, respectively). However, the size effect was smaller compared to their native compounds and significanlty different (*p* < 0.05) compared to the negative control. 

## 4. Discussion

In the present study, we documented the capacity of anthocyanins (Peo-3-glc and Pet-3-glc) to reduce the adhesion of monocytes to vascular endothelial cells, either when tested at physiological or supra-physiological concentrations. Conversely, the effect of their metabolites to counteract the adhesion of THP-1 to HUVECs was controversial. In particular, MetGA did not show any significant effect at each concentration tested, while VA was effective only at the maximum concentration. The present findings agree with our previous studies that reported the ability of an anthocyanin-rich fraction, single anthocyanins (cyanidin, delphinidin and malvidin-3-glucoside) and their relative metabolites (protocatechuic, gallic and syringic acid) to differentially prevent and/or resolve (depending on the compound and dose tested) an inflammatory response and mitigate the adhesion of monocytes to endothelial cells—an important initial step of the atherogenic process [26,27]. The ability of anthocyanins and metabolites to reduce/prevent the adhesion of monocytes/macrophages to endothelial cells has been reported in several studies, even if the results are not always in agreement with each other. This could be due to the different compounds and concentrations tested. Most of the studies reported in the literature used supra-physiological concentrations as it is well recognized that anthocyanins are scarcely absorbed [10]. Generally, their blood concentrations range from 0.06 nM up to 0.4 µM, while those of their metabolites range between 0.2 μM and 2 μM [10]. In our experimental conditions, we tested both plasma relevant concentrations (0.02 μM and 0.2 µM for anthocyanins and 0.05 μM, 0.5 μM and 5 µM for their metabolites) and supra-physiological (2 μM and 20 µM for anthocyanins and 50 µM for metabolites), supporting (at least in part) the role of physiological doses in the modulation of the adhesion process. However, the results obtained were dependent on the molecules used. The effect of anthocyanins and metabolites at plasma concentrations has been evaluated in few studies. For example, Krga and colleagues [28] tested the effects of 10 different phenolic compounds (five anthocyanins and five degradation products/gut metabolites) on the capacity to counteract the adhesion of monocytes to endothelial cells. The authors reported a significant reduction in the adhesion process following delphinidin-3-glucoside treatment at all the concentrations tested, cyanidin-3-glucoside, galactoside and arabinoside in the range between 0.1–2 µM, while Peo-3-glc was effective only at the lowest concentration. Considering anthocyanin metabolites, protocatechuic acid reduced monocyte adhesion at all concentration tested, VA at 0.2 μM and 2 µM only, while ferulic and hippuric acids were only effective at 1 μM and 2 µM [28]. The results obtained on Peo-3-glc and VA are only partially in line with our observations, since the effects of VA were detected only at the maximum dose (50 µM). In addition, we cannot exclude that other factors could have affected the findings obtained; for example, an important factor of variability may depend on the different experimental design adopted between the two studies. We tested Peo-3-glc and VA after an overnight stimulation with 100 ng mL^−1^ of TNF-α and a co-incubation with monocytes, while Krga and coworkers [28] incubated them for different times (3 h for Peo-3-glc e and 18 h for VA) and the stimulation with TNF-α was performed for 4 h while monocyte co-incubation was limited to 15 min.

The mechanisms of action through which polyphenols can reduce/prevent the adhesion process and consequently exert their anti-atherosclerotic effect are still not completely understood. It is widely recognized that atherosclerosis is a multifactorial process involving several pathways. It is also well-known that chronic inflammation may activate this process, starting with the over-expression and production of different cytokines, interleukins and adhesion molecules such E-selectin, VCAM-1 and ICAM-1. E-selectins are Ca^2+^-dependent transmembrane lectins, produced following different stimuli such as TNF-α, IL-1β and LPS, that permit the rolling of monocytes to endothelial cells. Moreover, this process enhances the expression of β2-integrin which allows the strong adhesion and the transmembrane migration of the monocytes at the endothelial level [29]. For this reason, E-selectin plays a major role and represents an important molecular target in the study of atherosclerosis. Together with E-selectin, VCAM-1 is also an important protein involved in the initiation of the atherosclerotic process. In fact, the activation of endothelial cells stimulates the expression of VCAM-1 which is able to bind α4β1 integrin located on monocyte membranes by determining the rolling-type adhesion and later the firm adhesion phase [30]. It has been observed that administration of monoclonal antibodies against VCAM-1 can reduce monocyte adhesion to endothelial cells and decrease plaque formation in apolipoprotein E-deficient (ApoE−/−) mice [31]. Few studies that examined the role of polyphenols on the modulation of E-selectin and VCAM-1 expression/production have documented different results depending on the type of compound tested. For example, Warner et al. reported that phenolic metabolites of different flavonoids, but not their unmetabolized precursors, were able to reduce the secretion of VCAM-1 at a range of concentrations between 1 μM and 100 µM [32]. Similar results were reported by Kunts and colleagues, showing that microbial fermentation of an anthocyanin-rich grape/berry extract (50 µM) reduced the expression of the adhesion molecules E-selectin, VCAM-1 and ICAM-1. However, this effect was dependent on bacterial species and is most likely due to their capacity to biotransform anthocyanins [33]. Amin et al. showed that the incubation of cyanidin-3-glucoside and different metabolites, in particular ferulic acid, at different concentrations (0.1 μM, 1 μM, and 10 µM) were able to alter the expression of VCAM-1 at physiologically relevant concentrations [34]. More recently, Calabriso et al. reported the capacity of a biofortified bread polyphenol extract (containing mainly ferulic, sinapic and p-coumaric acids) to inhibit in a concentration-dependent manner (1 μg mL^−1^, 5 μg mL^−1^, 10 μg mL^−1^) the adhesion of monocytes to LPS-stimulated endothelial cells through a reduction in the expression of different adhesion molecules, with a significant effect on VCAM-1 [35]. In our in vitro model, Peo-3-glc and Pet-3-glc significantly inhibited the production of E-selectin at all tested concentrations while VA was effective only at supra-physiological concentrations according to the results regarding the adhesion process. Differently, Peo-3-glc was the only compound able to decrease the levels of VCAM-1 at physiologically-relevant concentrations while no effect was observed for Pet-3-glc, VA and MetGA, confirming the results of our previous publication [26] and in line with results found by others researchers [28,36], suggesting that high concentrations are needed in order to exert an effect.

The different effects obtained with anthocyanins compared to their metabolites may be explained by their variable structures, chemical properties, and thus their heterogeneous capacity to interact with biological systems and to modulate target molecules. The presence of several functional groups, but also the size of the molecule or different conformations could be all factors affecting the binding of these compounds to specific membrane receptors, the interaction with transcriptional factors, or the capacity to act as a free-radical scavenger. Moreover, the potential synergistic role of phenolic compounds on the regulation of the main processes in which they are involved should also be taken into account.

The role of angiogenesis in atherosclerotic plaque progression is still not completely understood. Despite several in vitro studies showing that VEGF-induced angiogenetic processes increase plaque instability, the administration of anti-angiogenic drugs (mainly anti-VEGF) for cancer therapy causes adverse cardiovascular effects in human studies. A recent review asserts that the long-term treatment of oncological patients with anti-VEGF drugs could promote adverse cardiovascular effects through hypertension, suggesting a different mechanism of action of VEGF inhibitors compared to in vitro studies that aim to evaluate the role of angiogenesis within the plaque [37]. Neocapillaries inside the atherosclerotic plaque are more fragile and can easily undergo damage due to the high level of oxidative stress that mainly occurs during the later stage of atherosclerosis. This latter condition could lead to plaque rupture, one of the main factors responsible for cardiovascular events [4]. Arterial injuries are followed by arterio-intimal angiogenesis that induces intimal hyperplasia and a subsequent intimal hemorrhage [38]. Repeated intraplaque hemorrhages play an essential and promoting role in plaque progression and rupture. Intraplaque hemorrhages are mainly induced by angiogenesis from the adventitia to the intima, where the atheroma starts to develop [5]. To support the hypothesis of the involvement of angiogenesis in atherosclerosis, Qiu et al. showed that arterial regions with higher shear stress also exhibit an elevated number of intraplaque microvessels, characterized by abnormal endothelial cells, in particular with intracytoplasmatic vacuoles and leukocyte infiltration that could lead to rupture-prone plaque formation [39]. In cancer research, multiple in vitro studies demonstrated the anti-angiogenic effect of anthocyanins, in particular concerning delphinidin, as a potential chemopreventive agent [40,41,42]. We found that Peo-3-glc, Pet-3-glc and their metabolites (VA and MetGA) reduced the levels of VEGF, corroborating the hypothesis of a protective mechanism of action through which these compounds inhibit angiogenesis within the atheroma, therefore reducing atherosclerotic disease progression. Tanaka et al., using a purple rice extract and its constituents cyanidin and peonidin tested at 10 μg mL^−1^ and 30 μg mL^−1^ on HUVECs and HRMECs, showed a reduction of migration and proliferation. In detail, these polyphenols seem to act through the inhibition of extracellular signal-regulated kinase (ERK) 1/2 and p38 pathways in reducing VEGF-induced angiogenesis [43]. Similar results were observed by Negrao et al., who reported that 1 µM of catechin was able to reduce migration and invasion capacity in smooth muscle cells. This latter effect seems to depend on the presence or absence of angiogenesis stimuli, such as VEGF, emphasizing a potential use of some phenolic compounds against pathological situations where angiogenesis is stimulated [44]. Also, Calabriso et al. demonstrated that 0.1 μg mL^−1^ to 10 μg mL^−1^ of olive oil polyphenol extract suppressed endothelial cell migration induced by VEGF. The inhibition was dose-dependent, and the lowest concentration reduced the migration by about 35% [45]. For the first time, Tsakiroglou et al. reported a different modulation of endothelial cell migration through the regulation of ras homolog family member A (RHOA) and ras-related C3 botulinum toxin substrate (RAC)-1 (two proteins involved in cell motility), induced by anthocyanin and the phenolic fraction from wild blueberries (dependent on dose and compound). In detail, time-lapse videos showed that the anthocyanin fraction at 60 μg mL^−1^ decreased the migration rate of endothelial cells, while treatment with the phenolic acid fraction at 0.002 μg mL^−1^, 60 μg mL^−1^ and 120 μg mL^−1^ significantly increased the endothelial cell migration rate [46]. Cerezo et al. tested a wide range of polyphenols on VEGF-dependent vascular endothelial growth factor receptor (VEGFR)-2 activation. In particular, 11 of these phenolic compounds showed an half-maximal inhibitory concentration (IC50) < 1 µM, demonstrating efficacy at physiologically relevant concentrations. These compounds act by binding to a specific site of VEGF while avoiding the interaction with its receptor VEGFR2. The inhibitory potency is strongly correlated to the binding affinity that, in turn, is related to structural features such as the galloyl group at the 3-position of flavan-3-ols, the degree of polymerization of procyanidin oligomers, the total number of hydroxyl groups on the B-ring and hydroxylation of position 3 on C-ring [47]. In a subsequent study, Perez-Moral et al. reported that polyphenols with a strong inhibitory effect toward VEGF also have a lower IC50, demonstrating the increased formation of complexes between VEGF and polyphenols (and vice versa for those with a higher IC50), highlighting that the level of VEGF inhibition is strongly correlated to VEGF–polyphenol complex formation. To strengthen these results, polyphenols with lower IC50 values also demonstrated lower dissociation rate constants and equilibrium dissociation constants, indicating a stronger interaction and higher affinity [48]. A recent review reported that the anti-angiogenic role of anthocyanins is more consistent compared to phenolic acids, for which results are still mixed. According to Tsakiroglou et al., this heterogeneity is mainly due to the use of different types, combinations and concentrations of the compounds tested, but also to different cell lines, co-cultures and types of stimulation. Therefore, enhanced scientific cooperation, using common extracts and experimental protocols, could lead to a consensus among different studies, thereby formulating robust conclusions [49].

## 5. Conclusions 

Taken together, our results have shown that Peo-3-glc and Pet-3-glc, but not VA and MetGA, decrease the attachment of monocytes to endothelial cells via E-selectin reduction. These results were documented both at physiological and supra-physiological concentrations, providing further evidence of the capacity of polyphenols to blunt inflammation and to counteract the processes involved in the onset of atherosclerosis. Moreover, we documented (for the first time) the important role of Peo-3-glc and Pet-3-glc and their metabolites to reduce VEGF and thus exert an important role on the modulation of angiogenesis. Studies are ongoing in order to corroborate the findings obtained and to elucidate the contribution of these and other polyphenols (alone or in combination) in the modulation of further important molecules potentially involved in the adhesion process, such as intercellular adhesion molecules 1, L- and P-selectin, and endothelin-1.

## Figures and Tables

**Figure 1 nutrients-12-00655-f001:**
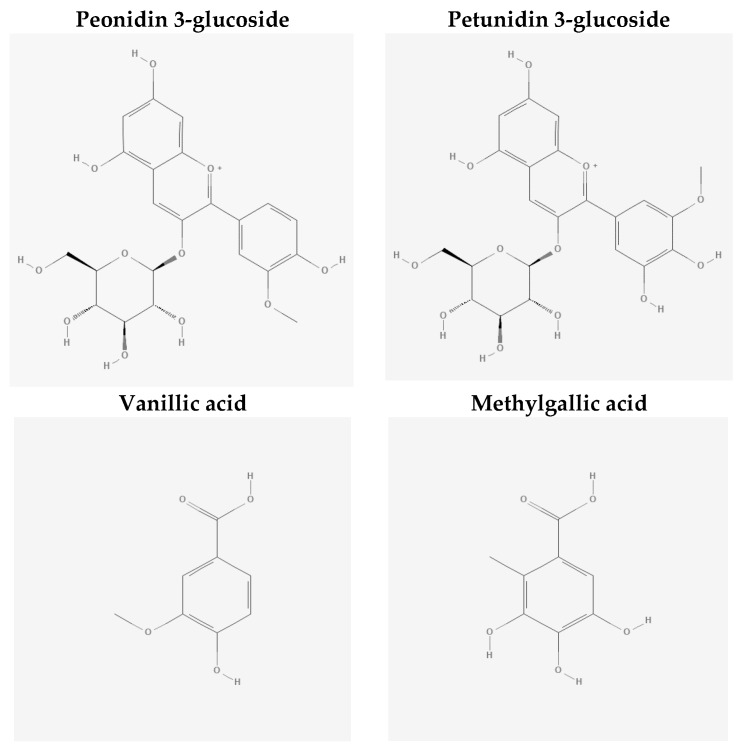
Chemical structure of peondin and petunidin-3-glucoside, vanillic and methylgallic acids.

**Figure 2 nutrients-12-00655-f002:**
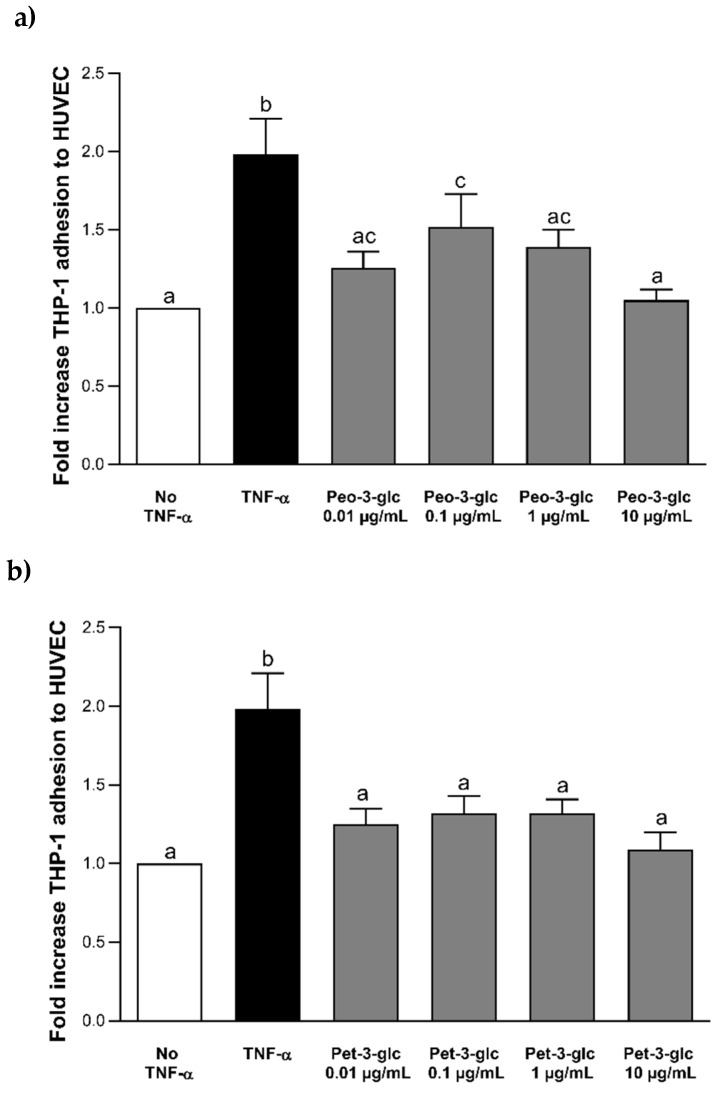
Effect of different concentrations (0.02–20 μM) of Peo-3-glc (**a**) and Pet-3-glc (**b**) on THP-1 (monocytes) adhesion to HUVECs (vascular endothelial cells). Results are expressed as mean ± standard error of mean. ^a,b,c^ Bar graphs reporting different letters are significantly different (*p* ≤ 0.05).

**Figure 3 nutrients-12-00655-f003:**
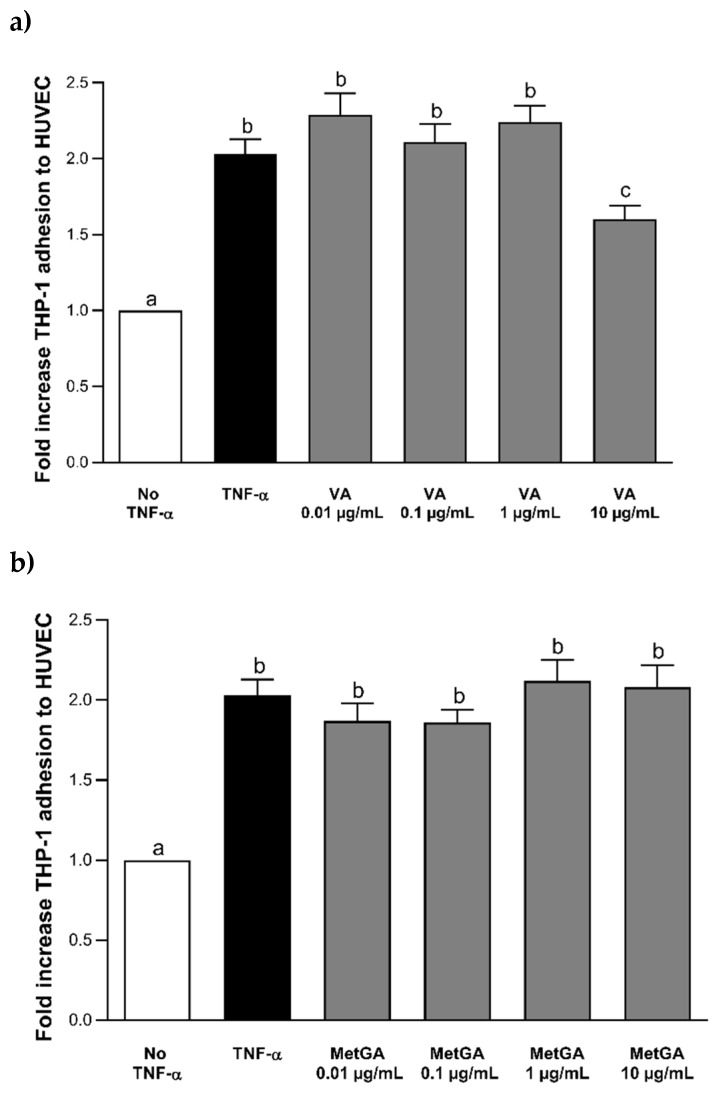
Effect of different concentrations (0.05–50 μM) of VA (**a**) and MetGA (**b**) on THP-1 adhesion to HUVECs. Results are expressed as mean ± standard error of mean. ^a,b,c^ Bar graphs reporting different letters are significantly different (*p* ≤ 0.05).

**Table 1 nutrients-12-00655-t001:** Percentage of cell viability following supplementation with peonidin-3-glucoside (Peo-3-glc), petunidin-3-glucoside (Pet-3-glc), vanillic acid (VA) and methyl-gallic acid (MetGA) evaluated by Trypan blue and MTT assays.

Trypan Blue Assay	Anthocyanins			Gut Metabolites	
Concentrations	Peo-3-glc	Pet-3-glc	Concentrations	VA	MetGA
0.02 µM	99.7 ± 0.33	110 ± 0	0.05 µM	100 ± 0	99.7 ± 0.33
0.2 µM	100 ± 0	97.0 ± 1.0	0.5 µM	99.7 ± 0.33	99.3 ± 0.67
2 µM	99.3 ± 0.67	97.7 ± 0.33	5 µM	99.7 ± 0.66	98.7 ± 1.33
20 µM	99.3 ± 0.33	100 ± 0	50 µM	99.3 ± 0.67	97.3 ± 1.77
**MTT assay**	**Anthocyanins**			**Gut metabolites**	
Concentration	Peo-3-glc	Pet-3-glc	Concentration	VA	MetGA
20 µM	98.5 ± 0.12	94.4 ± 0.45	50 µM	99.7 ± 0.32	96.7 ± 0.43

Results derived from three independent experiments. Peo-3-glc, Pet-3-glc, VA and MetGA were tested in the presence of tumor necrosis factor-α (TNF-α) stimulus. Each concentration was tested in triplicate. Data are reported as mean ± standard error of the mean.

**Table 2 nutrients-12-00655-t002:** Effect of peonidin-3-glucoside, petunidin-3-glucoside, vanillic acid and methyl-gallic acid on the levels of E-selectin.

	Anthocyanins			Gut Metabolites	
Concentrations	Peo-3-glc	Pet-3-glc	Concentrations	VA	MetGA
TNF-α 0 ng mL^−1^	160 ± 7.9 ^a^	164 ± 5.8 ^a^	TNF-α 0 ng mL^−1^	160 ± 7.9 ^a^	164 ± 5.8 ^a^
TNF-α 100 ng mL^−1^	316 ± 8.1 ^b^	317 ± 6.3 ^b^	TNF-α 100 ng mL^−1^	316 ± 8.1 ^b^	317 ± 6.3 ^b^
0.02 µM	143 ± 4.3 ^a^	115 ± 7.5 ^c^	0.05 µM	312 ± 14.1 ^b^	299 ± 7.5 ^b^
0.2 µM	108 ± 5.3 ^c^	123 ± 11.8 ^a,c^	0.5 µM	312 ± 11.2 ^b^	297 ± 7.5 ^b^
2 µM	109 ± 7.2 ^c^	104 ± 6.3 ^c^	5 µM	305 ± 7.4 ^b^	297 ± 8.0 ^b^
20 µM	76 ± 8.4 ^c^	88 ± 12.1 ^c^	50 µM	95 ± 13.2 ^c^	295 ± 7.3 ^b^

Results derived from three independent experiments. Peo-3-glc, Pet-3-glc, VA and MetGA were tested in the presence of TNF-α stimulus. Each concentration was tested in triplicate. Data are reported as mean ± standard error of the mean. ^a,b,c^ Data with different letters are significantly different (*p* < 0.05).

**Table 3 nutrients-12-00655-t003:** Effect of peonidin-3-glucoside, petunidin-3-glucoside, vanillic acid and methyl-gallic acid on the levels of sVCAM-1.

	Anthocyanins			Gut Metabolites	
Concentrations	Peo-3-glc	Pet-3-glc	Concentrations	VA	MetGA
TNF-α 0 ng mL^−1^	59 ± 9.0 ^a^	64 ± 10 ^a^	TNF-α 0 ng mL^−1^	59 ± 9.0 ^a^	64 ± 10 ^a^
TNF-α 100 ng mL^−1^	316 ± 16 ^b^	307 ± 11 ^b^	TNF-α 100 ng mL^−1^	316 ± 16 ^b^	307 ± 11 ^b^
0.02 µM	107 ± 15 ^c^	311 ± 13 ^b^	0.05 µM	308 ± 11 ^b^	299 ± 15 ^b^
0.2 µM	104 ± 16 ^c^	297 ± 15 ^b^	0.5 µM	299 ± 22 ^b^	297 ± 15 ^b^
2 µM	186 ± 12 ^c^	300 ± 14 ^b^	5 µM	295 ± 12 ^b^	297 ± 16 ^b^
20 µM	149 ± 24 ^c^	83 ± 10 ^c^	50 µM	315 ± 16 ^c^	295 ± 14 ^b^

Results derived from three independent experiments. Peo-3-glc, Pet-3-glc, VA and MetGA were tested in the presence of TNF-α stimulus. Each concentration was tested in triplicate. Data are reported as mean ± standard error of the mean (SEM). ^a,b,c^ Data with different letters are significantly different (*p* < 0.05).

**Table 4 nutrients-12-00655-t004:** Effect of peonidin-3-glucoside, petunidin-3-glucoside, vanillic acid and methyl-gallic acid on the levels of VEGF.

	Anthocyanins			Gut Metabolites	
Concentrations	Peo-3-glc	Pet-3-glc	Concentrations	VA	MetGA
TNF-α 0 ng mL^−1^	120 ± 6.9 ^a^	121 ± 6.1 ^a^	TNF-α 0 ng mL^−1^	120 ± 6.9 ^a^	121 ± 6.1 ^a^
TNF-α 100 ng mL^−1^	170 ± 8.5 ^b^	172 ± 7.9 ^b^	TNF-α 100 ng mL^−1^	170 ± 8.5 ^b^	172 ± 7.9 ^b^
0.02 µM	120 ± 6.9 ^a^	129 ± 10 ^a^	0.05 µM	149 ± 3.0 ^c^	153 ± 2.5 ^c^
0.2 µM	123 ± 1.7 ^a^	123 ± 7.4 ^a^	0.5 µM	141 ± 8.3 ^c^	142 ± 3.0 ^c^
2 µM	123 ± 6.0 ^a^	123 ± 2.9 ^a^	5 µM	147 ± 4.9 ^c^	141 ± 4.9 ^c^
20 µM	119 ± 2.6 ^a^	117 ± 9.9 ^a^	50 µM	135 ± 5.7 ^c^	137 ± 6.0 ^c^

Results derived from three independent experiments. Peo-3-glc, Pet-3-glc, VA and MetGA were tested in the presence of TNF-α stimulus. Each concentration was tested in triplicate. Data are reported as mean ± standard error of the mean (SEM). ^a,b,c^ Data with different letters are significantly different (*p* < 0.05).

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
