# Peer review of "Modulation of Adhesion Process, E-Selectin and VEGF Production by Anthocyanins and Their Metabolites in an In Vitro Model of Atherosclerosis"

_nutrients, 2020, doi:10.3390/nu12030655_

Round 1

Reviewer 1 Report

In the manuscript submitted by Marino et al, the authors assessed the capacity of metabolites to decrease monocyte adhesion to endothelial cells as well as secretion of cell adhesion molecules. The manuscript is well written, and experiments are clear. However, there are important points that authors must address:

-The authors should give more information about pharmacokinetics of these molecules and their metabolites, at what concentrations are these compounds found in the circulation, for how long.

-The authors need to compare concentrations used and those that had an effect with concentrations found in the circulation, are they achievable? And duration of 24h? 50microM is high concentration. These points need to be added in the introduction and discussion.

-Resolution of the image is low and difficult to read

-For figure: “noTNF” bar could be on the left next to “TNF” bar and easier to see the effect of TNF rather than leave it on the right

-Also for the tables, the authors should put the negative control, NO TNF, first then TNF and then the molecules.

Author Response

Question 1: The authors should give more information about pharmacokinetics of these molecules and their metabolites, at what concentrations are these compounds found in the circulation, for how long.

Answer: We agree with the reviewer and added information in the introduction as requested

Question 2: The authors need to compare concentrations used and those that had an effect with concentrations found in the circulation, are they achievable? And duration of 24h? 50microM is high concentration. These points need to be added in the introduction and discussion.

Answer: We thank the reviewer for the right comments. We tried to improve introduction and the discussion as requested. 

Question 3: Resolution of the image is low and difficult to read

Answer: All the figures were provided in high resolution and uploaded also separately from the main manuscript. Unfortunately, the images lost their resolution during the copy and paste in the word format. We tried to improve their quality.

Question 4: For figure: “noTNF” bar could be on the left next to “TNF” bar and easier to see the effect of TNF rather than leave it on the right

Answer: We changed as suggested. Thank you

Question 5: Also for the tables, the authors should put the negative control, NO TNF, first then TNF and then the molecules

Answer: We modified accordingly.

Reviewer 2 Report

The manuscript is generally well writen and clearly presented. However, I made some suggestions:

Introduction:

I recommend adding the main plant sources  for peonidin and petunidin.

L70: I recommend changing the expression:... atheroprotective properties...

Reviewer 3 Report

The authors investigated the ability of peonidin, petunidin-3-glucoside, vanillic acid and methyl-gallic acid to prevent monocyte adhesion to endothelial cells and measured their influence on the production of VCAM-1, E-selectin and VEGF. They found that anthocyanins significantly reduced monocyte adhesion while their metabolites did not (except for the highest concentration of vanillic acid). All of the tested substances decreased VEGF production. The manuscript is well-written and the results are clearly presented.

Minor comments:

  • Did the authors think about measuring additional adhesion molecules like e.g. ICAM-1, that has been widely studied with regard to smoking?
  • Could the anthocyanins also influence cell adhesion molecules on the monocytes, e.g. L-selectin? Did the authors check that?
  • Line 247: Typing error: “selectin, VCAM-1 and ICAM-1. E-selectins are a Ca2+-dependent transmembrane lectins,…”

Author Response

Question 1: Did the authors think about measuring additional adhesion molecules like e.g. ICAM-1, that has been widely studied with regard to smoking?

Answer: We thank the reviewer for the comment. In present study, we did not measure the levels of ICAM-1 but we are performing studies in which also the contribution of ICAM-1 and other important cytokines/interleukins are considered. We added a sentence in the conclusions.   

Question 2: Could the anthocyanins also influence cell adhesion molecules on the monocytes, e.g. L-selectin? Did the authors check that?

Answer: We thank the reviewer for this very interesting observation. We know that L-selectin is a transmembrane glycoprotein widely expressed on most circulating leukocytes especially during the inflammation process. Some studies reported the capacity of polyphenols (in general) to affect the expression of this important adhesion molecule; in the present paper, we did not measure its production. However, we are conducting studies and its evaluation, together with other markers, is under consideration.

Question 3: Line 247: Typing error: “selectin, VCAM-1 and ICAM-1. E-selectins are a Ca2+-dependent transmembrane lectins,…”

Answer: We apologize with the reviewer. The manuscript has been revised for the language and also for typing errors.